# Experimental Study on the Application of Recycled Concrete Waste Powder in Alkali-Activated Foamed Concrete

**DOI:** 10.3390/ma16175728

**Published:** 2023-08-22

**Authors:** Dongsheng Zhang, Weiwei Hao, Qiuning Yang

**Affiliations:** 1School of Civil and Hydraulic Engineering, Ningxia University, Yinchuan 750021, China; dongsheng.zhang@kuleuven.be (D.Z.); chenwu234@foxmail.com (W.H.); 2Research Group RecyCon, Department of Civil Engineering, KU Leuven, Campus Bruges, 8200 Bruges, Belgium

**Keywords:** recycled concrete powder, foamed concrete, strength, drying shrinkage, frost resistance

## Abstract

The alkali-activated cementitious material was prepared by partially replacing slag with recycled concrete powder (RCP). The influence of RCP substitution rates (10%, 20%, 30%, 40%, and 50% mass fraction) on the performance of alkali-activated slag-RCP-based (AASR) foamed concrete was studied. The fluidity, water absorption, softening coefficient, compressive strength, flexural strength, drying shrinkage, thermal conductivity, and frost resistance of AASR foamed concrete were studied. The results show that the fluidity and softening coefficient of AASR foamed concrete decreases with the increase in RCP content, and the fluidity range is 230–270 mm. Due to the porous structure of the RCP, the water absorption of AASR increases. With the increase in the curing age, the strength of AASR foamed concrete increases. The addition of RCP reduced the mechanical properties of AASR foamed concrete. Compared with the control group, the compressive strength of AASR50 decreased by 66.7% at 28 days, and the flexural strength decreased by 61.5%. However, the 28 d compressive strength of AASR foamed concrete under all RCP replacement rates still meets the standard value (0.6 MPa). The addition of RCP effectively reduces the thermal conductivity of the AASR foamed concrete, and when the RCP content is 50%, the thermal conductivity is lowest, 0.119 W/(m·K); the drying shrinkage of the AASR foamed concrete can be improved by adding RCP, and the drying shrinkage value is lowest when the RCP is 30%, which is 14.7% lower than that of the control group. The frost resistance of AASR foamed concrete decreases with the increase in the RCP content. When the recycled micropowder content is 20–50% and after 25 freeze–thaw cycles, AASR foamed concrete has reached freeze–thaw damage.

## 1. Introduction

Foamed concrete is a kind of lightweight inorganic material, which has a wide range of application scenarios and can be used as filling materials, insulation boards, lightweight partitions, etc. [1]. Compared with traditional insulation filling materials, foamed concrete has the characteristics of low cost, controllable density, convenient construction, etc. [2,3]. Widespread application of foamed concrete cannot only effectively reduce the dead weight of buildings, reduce the production, manufacturing, and labor costs involved in the preparation and transportation processes, but also greatly reduce energy consumption [4,5,6]. For a long time, Portland cement has mostly been used as the cementing material, and the foamed concrete prepared by this method has the characteristics of excellent durability and easy construction. However, during the preparation process of Portland cement, massive energy is consumed, which goes against the modern concept of energy conservation and emission reduction. In addition, Portland cement has a long setting time, which will lead to partial defoaming during the preparation of foamed concrete, reducing the volume of foamed concrete and even collapsing the mold [7,8].

Alkali-activated slag-foamed concrete emerged to solve this issue. Since it does not need cement in the production process, it is a low-carbon and environmentally friendly material, with the advantages of high early strength and good water resistance [9,10,11]. The hydration product of the high calcium cementitious system represented by slag is mainly hydrated calcium aluminum silicate (C-A-S-H). Its reaction process has a lower hydration heat and faster hydration rate than traditional cement, making it a promising alternative material for Portland cement [12,13,14]. Su et al. [15] used fly ash and slag as the main raw materials to prepare foamed concrete with the alkali-activation method. It was found that when the amount of activator was 21% and the modulus of water glass was one, foamed concrete showed the best performance. After 28 days, the compressive strength of the sample reached 2.18 MPa, and the porosity was 63.07%. The compressive strength of alkali-activated slag foamed concrete with a density of 600–1500 kg/m^3^ prepared by Mastali et al. [16] is 25–130 MPa; compared with ordinary Portland cement-based foamed concrete with the same density, the strength is higher. He et al. [17] showed that the water absorption rate of alkali-activated slag foamed concrete with a dry density of 600 kg/m^3^ was between 50.1 and 52.6%, with good durability. However, despite China producing over 100 million tons of slag annually, the market demand is high, and slag prices have been rising, and the continued high demand may limit future slag supplies [18]. Therefore, it is necessary to search for materials that can partially replace slag in alkali-activated cementitious materials.

With the acceleration of urbanization, massive construction waste, mainly concrete, has been produced. How to deal with these wastes has become an important issue for the sustainable development of buildings. Nowadays, many enterprises and scholars use crushed construction waste to produce aggregates [19,20,21,22] to replace increasingly scarce natural stone resources. However, in the process of producing recycled concrete aggregates, crushing, screening, and other steps will generate a large number of particles smaller than 75 microns, which are called recycled concrete powder (RCP). Recycling of RCP is an important part of achieving zero emissions of construction waste [19,23,24,25]. In recent years, the use of RCP as a partial replacement for slag and as a cementitious material to prepare alkali-activated cement has received a lot of attention. Sharma [26] used sodium hydroxide and sodium metasilicate solution as an activator to study the preparation of geopolymer mortar by partially replacing fly ash with recycled micropowder. It was found that the replacement rate of RCP reached 30%, which can improve the strength and durability of the geopolymer mortar. Additionally, the existence of calcium content in RCP was verified in microstructure analysis, which made the microstructure dense and improved the various properties of the geopolymer mortar. Liu [27] studied the drying shrinkage performance of the RCP-slab-based geopolymer (RPSG), and the results showed that slag can significantly reduce the drying shrinkage of RPSG, and the drying shrinkage rate of RPSG increases with the increase in alkali equivalent. Liu [28] reported on the microscopic characteristics, mechanical strength, and transportation performance of the alkali-activated slag geopolymers with different RCP contents. The results show that RCP in various construction wastes mainly consists of hydrated products, anhydrous cement particles, quartz, and calcite. The addition of high-content RCP expanded the pore structure of the slag-based polymer, and the replacement of 100% slag with RCP resulted in a relatively loose microstructure of a sustainable geopolymer. The mechanical strength and water resistance of the slag-based polymer decrease with the increase in the RCP substitution rate, and the replacement of 100% slag with RCP results in a significant decrease in the mechanical strength and water absorption. It is feasible and sustainable to use 100% RCP for upcycling of fully recovered geopolymer. Li [29] studied the influence and mechanism of different RCP substitution rates on mortar from the aspects of physical properties, mechanical properties, hydration kinetics, hydration products, pore structure, etc. The results showed that the addition of regenerated micropowder shortened the setting time of the slurry. With the increase in the RCP content, the hydration induction period of mortar is prolonged, and the peak hydration heat release rate and cumulative heat release show a significant downward trend. Obviously, it is feasible to use RCP as an economical and environmentally friendly cementitious material to replace some slag in the alkali slag cementitious system. However, there are a few reports on the research on the preparation of AASR foamed concrete with RCP. Therefore, the purpose of this experimental study is to explore the influence of RCP with different contents on the performance of alkali-activated slag foamed concrete.

In this experimental scheme, AASR foamed concrete was prepared by adding RCP with different contents (0%, 10%, 20%, 30%, 40%, and 50%) to replace part of the slag, and the fluidity, water absorption, softening coefficient, compressive strength, flexural strength, drying shrinkage, and frost resistance of the AASR foamed concrete are explored. The results of this study contribute to expanding the application range of RCP and promoting the development of green cementitious materials.

## 2. Experimental Details

### 2.1. Raw Materials

The cementitious materials used in this study are slag and RCP. The slag is selected as S95 grade slag, with a specific surface area of 524 m^2^/kg and a density of 2780 kg/m^3^. The raw material for RCP is obtained from waste concrete collected during the demolition process of a reinforced concrete building in Ningxia. After crushing the waste concrete, it is preliminarily screened through a 1.18 mm square hole sieve, and the obtained material is ground into micropowder using a planetary mill. The chemical composition of slag and RCP is shown in Table 1.

The alkaline activator is made by mixing water glass and industrial flake sodium hydroxide with an initial modulus of 3.3. The foaming agent is a Legao foaming agent produced by Shandong Lego Company, mixed with tap water in a mass ratio of 1:50.

### 2.2. Mix Proportions, Casting, and Curing

At the beginning of the test, it was determined that the AASR foamed concrete with a target dry density of 500 kg/m^3^ was prepared by using RCP and slag as precursors. The water-binder ratio of foamed concrete is 0.34, the modulus of the activator is 1.0, and the alkali equivalent is 5%. RCP is used as a supplementary cementitious material to replace 10%, 20%, 30%, 40%, and 50% of the slag. Additionally, the specific mix design is shown in Table 2.

First, the slag and RCP are evenly mixed in a mixer. Then, water and an alkali activator are poured into the mixer and stirred for three minutes to obtain a slurry with uniform flow. Next, the prepared foamed mix is added to the slurry quickly and evenly. This is subsequently poured into the mold after mixing uniformly, wrapped with a plastic film, and placed in a standard curing room. This can be demolded after 24 h. Next, the demolded test block must be cured until the measurement age.

### 2.3. Test Method

According to standards [30,31,32], the fluidity, dry density, water absorption, and softening coefficient of AASR foamed concrete were determined. As shown in Figure 1, with reference to JG/T 266-2011 [30] and GB/T 17671-2021 [33], the compressive strength of the AASR foamed concrete with a size of 100 mm × 100 mm × 100 mm and the flexural strength of the AASR foamed concrete with a size of 40 mm×40 mm×160 mm are measured with a YAW-2000 electro-hydraulic servo pressure testing machine. With reference to GB/T10294 [32], the ADH3030 thermal conductivity tester (as shown in Figure 2) is used to measure the thermal conductivity of the AASR foamed concrete with a size of 300 mm × 300 mm × 30 mm. An SP-175 vertical shrinkage dilatometer (as shown in Figure 3) was used to measure the 3, 7, 14, 21, 28, and 56 d dry shrinkage values of the AASR foamed concrete with a size of 40 mm × 40 mm × 160 mm.

Ultrasonic testing is an effective non-destructive testing method. The HU-U81 concrete ultrasonic detector was used to measure the wave velocity of the AARS foamed concrete before and after freezing and thawing, as shown in Figure 4. The relative dynamic elastic modulus of the concrete specimen was calculated according to Equations (1) and (2):(1)$$E=\frac{\rho (1+V)(1-2V)}{1-V}v^{2}$$
where, E is the dynamic elastic modulus of the material; V is the Poisson’s ratio of the material; *v* is the ultrasonic wave velocity of the material; and ρ is the density of the material. Due to the small variations in the Poisson and density of concrete materials [34,35], the relative dynamic elastic modulus of foamed concrete can be calculated using the following equation:(2)$$E_{r}=\frac{v_{n}^{2}}{v_{0}^{2}}$$
where E_r_ is the relative dynamic elastic modulus of the specimen; V_n_ is the ultrasonic wave velocity of the specimen after n freeze–thaw cycles; and V_0_ is the initial wave velocity of the specimen before the freeze–thaw cycle.

## 3. Results and Analyses

### 3.1. Fluidity

The fluidity results of the AASR foamed concrete under different RCP replacement rates are shown in Figure 5. With the increase in the RCP content, the fluidity of the AASR foamed concrete decreases gradually. Compared with the fluidity of AASR0 (270 mm), the fluidity of AASR10, AASR20, AASR30, AASR40, and AASR50 decreases by 4.4%, 7.4%, 10.0%, 13.0%, and 14.8%, respectively. This is mainly because the surface of the RCP is rough and there are numerous interconnected voids inside. The addition of RCP will increase the water absorption capacity in the foamed concrete slurry, resulting in excessive internal friction resistance of the slurry, thus reducing the fluidity of the slurry [36,37].

### 3.2. Water Absorption

Figure 6 shows the influence of the RCP content on the water absorption of the AASR foamed concrete. It can be seen from Figure 6 that the water absorption of the AASR foamed concrete gradually increases with the increase in the RCP content. When the RCP content is 10%, 20%, 30%, 40%, and 50%, the water absorption of the AASR foamed concrete is 3.5%, 11.8%, 15.3%, 24.1%, and 26.6% higher than that of AASR0. This is because the porous structure of RCP will increase the capillary effect of the AASR foamed concrete in water [38]; in addition, the activity of RCP is lower than that of slag. With the increase in its content, the amount of the hydration products of the AASR foamed concrete decreases, and the compactness decreases [39].

### 3.3. Softening Coefficient

The softening coefficient is an important parameter for the water resistance properties of materials. It can be seen from Figure 7 that the softening coefficient of the AASR foamed concrete gradually decreases with the increase in the RCP content. When the content of RCP increases from 0% to 50%, the softening coefficient of the AASR foamed concrete decreases from 0.963 to 0.710, indicating that adding RCP will have a great impact on its mechanical properties after water absorption; thus, the content of RCP should be reasonably selected according to the actual conditions. The softening of the AASR foamed concrete after water absorption is mainly caused by the addition of RCP, which makes the concrete produce a certain amount of N-A-S-H gel after hydration. The gel structure is prone to hydrolysis due to unstable silicoaluminate bonds, which will cause the overall deterioration in the hardened cement paste, increase the micropores, and then lead to a decline in the macro strength. In addition, when AASR foamed concrete is saturated with water, the moisture migration in the internal pores will lead to a reduction of cohesive forces between crystal particles in the hardened cement paste and poor connection performance [40]. Finally, when AASR foamed concrete is subjected to external forces, the stress generated by the forced migration of internal moisture will cause certain damage to its pore structure, thereby reducing the strength of the AASR foamed concrete.

### 3.4. Compressive Strength

Figure 8 shows the influence of RCP content on the compressive strength of the AASR foamed concrete. With the increase in the curing age, the compressive strength of the AASR foamed concrete increased, especially in the later ages. At all ages, the compressive strength of foamed concrete decreases with the increase in the RCP content. When 50% RCP is added, the 3 d, 7 d, and 28 d compressive strength of the AASR foamed concrete is 0.47 MPa, 0.70 MPa, and 1.11 MPa, respectively, which is 66.7%, 63.2%, and 53.8% lower than that of AASR0. This is mainly because as the content of RCP increases, its disadvantages of having more internal pores, poor particle shape, and low activity gradually become prominent. Moreover, the more the content is added, the more water is adsorbed, which is not conducive to the hydration reaction; secondly, RCP replaces mineral powder. Because the activity of RCP is lower than that of slag, the compressive strength of the AASR foamed concrete decreases [41]. Although the addition of RCP reduces the compressive strength of the AASR foamed concrete, the 28 d compressive strength of AASR foamed concrete under all RCP replacement rates still meets the standard value (0.6 MPa).

### 3.5. Flexural Strength

The influence of the RCP content on the flexural strength of the alkali-activated foamed concrete is shown in Figure 9. With the increase in the RCP content, the flexural strength of AASR foamed concrete decreases gradually, which is consistent with the change rule of compressive strength. When 50% RCP is added, the 3 d, 7 d, and 28 d flexural strength of the AASR50 foamed concrete is 0.2 MPa, 0.3 MPa, and 0.49 MPa, respectively, which is 64.2%, 63.2%, and 66.7% lower than that of the AASR0.

The relationship between the compressive strength (*f_t_*) and flexural strength (*f_f_*) of the foamed concrete is complex, which is affected by the composition, density, pore structure, curing conditions, and curing age of the foamed concrete. Generally, there is a positive correlation between the compressive strength and the flexural strength of the foamed concrete. According to the test results, the relationship between the AASR foamed concrete flexural strength and the cube compressive strength is shown in Figure 10. Through regression, the relationship formula can be obtained as follows:(3)$$f_{f}=0.493x-0.06$$

It should be stated that the proposed equations with high accuracy (R^2^ = 0.958) have good agreements with the experimental outcomes.

### 3.6. Thermal Conductivity

Figure 11 shows the influence of different RCP contents on the thermal conductivity of the AASR foamed concrete. It can be seen from Figure 11 that the addition of RCP can effectively reduce the thermal conductivity of the AASR foamed concrete. In particular, the larger the amount of RCP, the more obvious the downward trend is. When the amount of RCP is 50%, the thermal conductivity of the AASR foamed concrete is the lowest, 0.119 W/(m·K). This is because RCP has irregular particles and high-water absorption capacity compared with slag, which will cause damage to the slurry foamed liquid film, leading to the increase in irregular large holes in the specimen and a decrease in the pore structure compactness. With the increase in the RCP replacement rate, the porosity of AASR foamed concrete increases, which is more obvious. At this point, due to the reduction in the slag, the hydration products decrease, the proportion of stable internal and external solid structures decreases, the efficiency of heat conduction decreases, and the thermal conductivity decreases [40].

### 3.7. Drying Shrinkage

The drying shrinkage values of the AASR foamed concrete with different RCP contents are shown in Figure 12. It can be found that the drying shrinkage of the AASR foamed concrete is divided into three stages. The first stage is the slow growth stage (3–7 d). At this time, the internal water content of the structure is high, and the water is lost along the relatively large connecting pores and capillary pores. This situation will cause changes in the capillary tension and surface free energy of the AASR foamed concrete, making the specimen gradually shrink [42]; simultaneously, at this stage, the substitution rates of RCP have no significant effect on the dry shrinkage of the AASR foamed concrete. When the RCP substitution rates are 0%, 10%, 20%, 30%, 40%, and 50%, the 7 d dry shrinkage values of the samples are 1.584, 1.492, 1.404, 1.322, 1.444, and 1.607 × 10^−6^ m, respectively. The second stage is the rapid growth stage (7–28 d); at this time, the water content of the specimen is largely lost under the dry environment and internal hydration, and the nanopores in the AASR foamed concrete lose water seriously and the gel begins to densify. The combined effects of the two led to the sharp shrinkage of the specimen volume [43]; in addition, with the increase in the RCP content, the drying shrinkage of the foamed concrete decreases first and then increases. The RCP content is 10%, 20%, 30%, and 40%, and the 28 d dry shrinkage values of the samples are 3.429, 3.242, 3.062, 2.893, 3.145 and 3.503 × 10^−6^ m, respectively, and the drying shrinkage performance of the AASR foamed concrete is the best under the 30% replacement rate of RCP. This is because as the substitution rate of RCP for mineral powder increases, the activity of RCP is low, the hydration reaction is slow, and the hydration products produced by hydration are reduced, resulting in smaller shrinkage [44]. In the third stage, after 28 d, the dry shrinkage value of the specimen slowly increases and tends to stabilize. Compared with the 28 d dry shrinkage values, the 42 d shrinkage values for AASR0, AASR20, AASR30, AASR40, and AASR50 increased by 5.86%, 5.91%, 5.88%, 6.17%, and 6.59%, respectively. Compared with the 42 d shrinkage values, the 56 d shrinkage values for AASR0, AASR20, AASR30, AASR40, and AASR50 increased by 2.32%, 2.86%, 3.39%, 3.33%, and 4.69%, respectively.

### 3.8. Frost Resistance

#### 3.8.1. Mass Loss

The mass loss rate is one of the important indicators reflecting the frost resistance of cement-based materials. Figure 13 shows that when the number of freeze–thaw cycles is low, the influence of the dosage on the loss rate is relatively small. And when the number of freeze–thaw cycles is five, the mass loss rate of AASR0 is the lowest at 1.33%. With the increase in the freeze–thaw cycles, the mass loss rate of the AASR foamed concrete under different RCP content increases, and with the increase in the RCP content, the increase is larger. This is because with the increase in the RCP content, the water absorption rate of the AASR foamed concrete increases and the softening coefficient decreases, which makes the specimens with higher RCP content subject to greater frost heave stress and more serious surface damage under the condition of multiple freeze–thaw cycles. The mass loss rates of AASR0, AASR10, AASR20, AASR30, AASR40, and AASR50 are 4.32%, 4.89%, 5.74%, 6.65%, 7.68%, and 8.72%, respectively, after 25 freeze–thaw cycles. According to JGJ/T 341-2014 [31], when the mass loss rate of the AASR foamed concrete with 20–50% RCP content reaches 5% after 25 freeze–thaw cycles, it can be considered that it has been damaged by freeze–thaw.

#### 3.8.2. Compressive Strength Loss

Figure 14 shows the influence of the RCP content on the strength loss rate of AASR foamed concrete under different freeze–thaw cycles. It can be seen from Figure 14 that when the RCP content is unchanged, the strength loss rate of the AASR foamed concrete increases with the increase in the freeze–thaw cycles; when the number of freeze–thaw cycles is constant, the strength loss rate increases with the increase in the RCP content and shows an increasing trend under different freeze–thaw cycles. When the RCP content is 50%, the highest strength loss rate is 19.88% after 25 freeze–thaw cycles. This is because the activity of RCP is low, and its incorporation will reduce the strength of the AASR foamed concrete, and the ability to resist the frost heaving stress caused by freezing and thawing will become weak and vulnerable to damage. In addition, the incorporation of RCP will increase the water absorption in the AASR foamed concrete, which will subject it to greater frost heave stress during a freeze–thaw cycle, thus causing more serious internal damage [45].

#### 3.8.3. Relative Dynamic Modulus of Elasticity

Figure 15 shows the influence of freeze–thaw cycles on the relative dynamic elastic modulus of the AASR foamed concrete with different RCP contents. It can be seen from Figure 15 that with the increase in the freeze–thaw cycles, the relative dynamic elastic modulus of the AASR foamed concrete gradually decreases, which indicates that freeze–thaw will aggravate the internal damage of the AASR foamed concrete, because the expansion of water freezing in the pores will increase the water pressure in the pores until the tensile strength is reached and the internal pores of the test block freeze and crack. Under the same freeze–thaw cycles, the relative dynamic elastic modulus of AASR foamed concrete decreases with the increase in the RCP content. The relative dynamic elastic modulus of AASR0, AASR10, AASR20, AASR30, AASR40, and AASR50 are 70.8%, 66.3%, 48.5%, 40.2%, 30.8%, and 24.64%, respectively, after 25 freeze–thaw cycles. According to JGJ/T 341-2014 [31], the relative dynamic elastic modulus of the AASR foamed concrete with 20–50% RCP content has decreased to less than 60% after 25 freeze–thaw cycles, which can be regarded as having been damaged by freeze–thaw, which is consistent with the result of mass loss.

## 4. Conclusions

In this study, RCP and slag are used as the main cementitious materials to prepare AASR foamed concrete with a density of 500 kg/m^3^. The main conclusions are as follows:(1)The fluidity and softening coefficient of the AASR foamed concrete decrease with the increase in the RCP content and its fluidity ranges from 230 mm to 270 mm. Due to the porous structure of the RCP, the water absorption of the AASR foamed concrete increased.(2)With the increase in the curing age, the strength of the AASR foamed concrete increases. The addition of RCP reduces the mechanical properties of the AASR foamed concrete; in particular, the larger the amount of RCP, the more obvious the downward trend. Although the addition of RCP reduces the compressive strength of the AASR foamed concrete, the 28 d compressive strength of the AASR foamed concrete under all RCP replacement rates still meets the standard value (0.6 MPa). According to the test results, this paper establishes a linear relationship between the compressive strength and flexural strength of the AASR foamed concrete under different RCP replacement rates, and the two have a high correlation.(3)The addition of RCP effectively reduced the thermal conductivity of the AASR foamed concrete, and when the RCP content was 50%, the thermal conductivity was the lowest, 0.119 W/(m·K); this is because RCP has irregular particles and high-water absorption capacity compared with slag, which will cause damage to the slurry foamed liquid film, leading to the increase in the irregular large holes in the specimen and the decrease in the hole wall compactness. With the increase in the RCP replacement rate, the porosity of the AASR foamed concrete increases, which is more obvious. At this point, due to the reduction in the slag, the hydration products decrease, the proportion of stable internal and external solid structures decreases, the efficiency of heat conduction decreases, and the thermal conductivity decreases.(4)The drying shrinkage of AASG foamed concrete can be improved by adding RCP, and the 28 d drying shrinkage value is the lowest when the amount of RCP is 30%, which is 15.6% lower than that of the control group. However, the drying shrinkage of AASR foamed concrete will increase slightly when the amount of RCP is too high, but it is still lower than the reference group; the frost resistance of the AASR foamed concrete decreases with the increase in the RCP content. When the recycled micropowder content exceeds 20%, after 25 freeze–thaw cycles, the mass damage rate of the AASR foamed concrete exceeds 5%, and the relative dynamic modulus of elasticity exceeds 60%, all of which have reached destruction.

This study primarily focused on the influence of the RCP replacement rate on the fluidity, water absorption, softening coefficient, compressive strength, flexural strength, thermal conductivity, drying shrinkage, and frost resistance of AASR foamed concrete. However, high-temperature resistance, carbonation, sulfate attack, and durability are also important factors affecting the performance of foamed concrete. Therefore, in future research, exploring the durability of AASR foamed concrete is necessary. Moreover, additional research is needed to improve the mechanical properties of the concrete due to the addition of RCP by utilizing fibers or nanomaterials.

## Figures and Tables

**Figure 1 materials-16-05728-f001:**
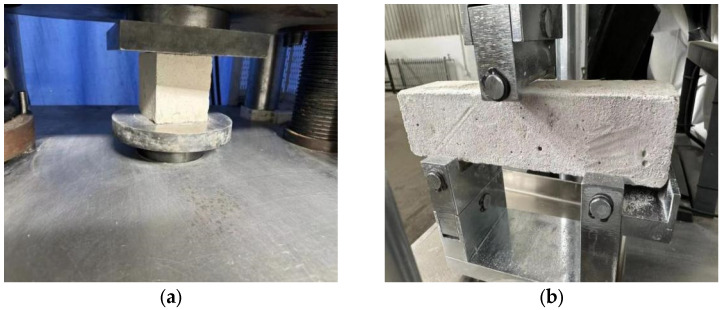
Measurement of mechanical properties of AASR foamed concrete, (**a**) Compressive strength test; (**b**) Flexural strength test.

**Figure 2 materials-16-05728-f002:**
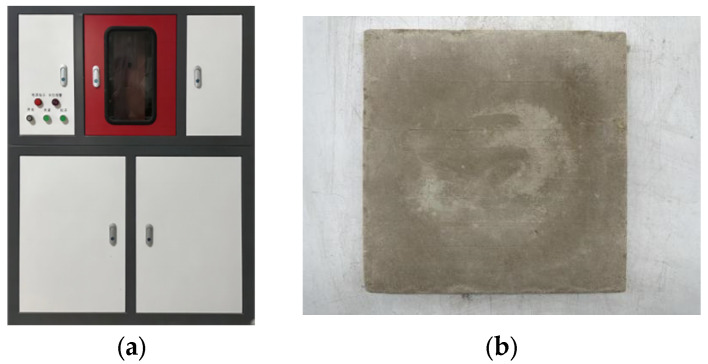
Thermal conductivity test of AASR foamed concrete, (**a**) Testing instrument; (**b**) The geometric shape of the sample.

**Figure 3 materials-16-05728-f003:**
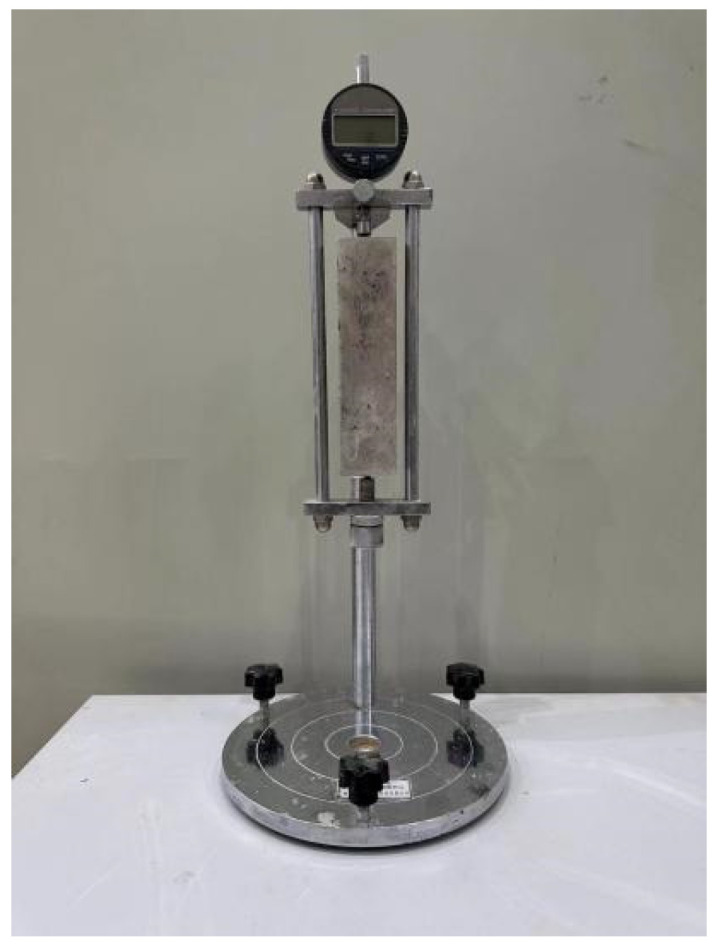
Drying shrinkage test of AASR foamed concrete.

**Figure 4 materials-16-05728-f004:**
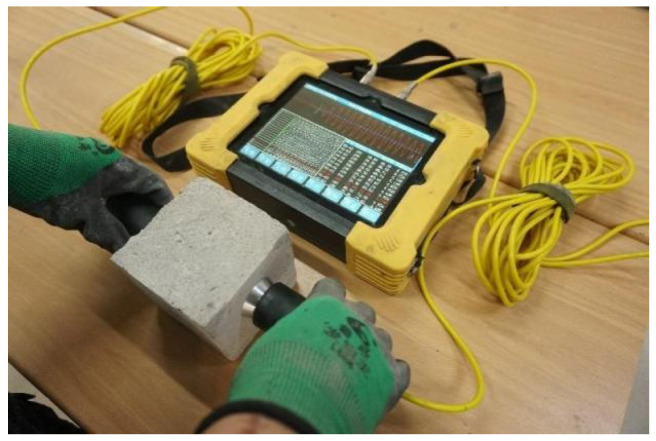
Ultrasonic wave velocity test of AASR foamed concrete.

**Figure 5 materials-16-05728-f005:**
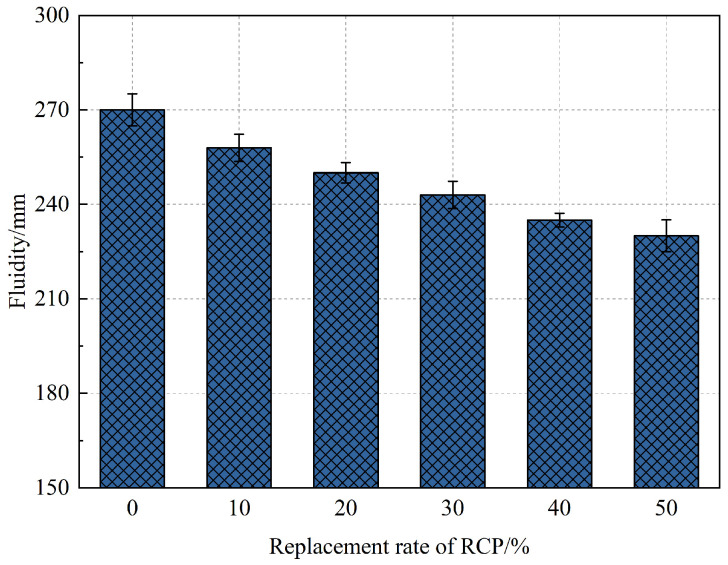
Effect of RCP contents on fluidity of AASR foamed concrete.

**Figure 6 materials-16-05728-f006:**
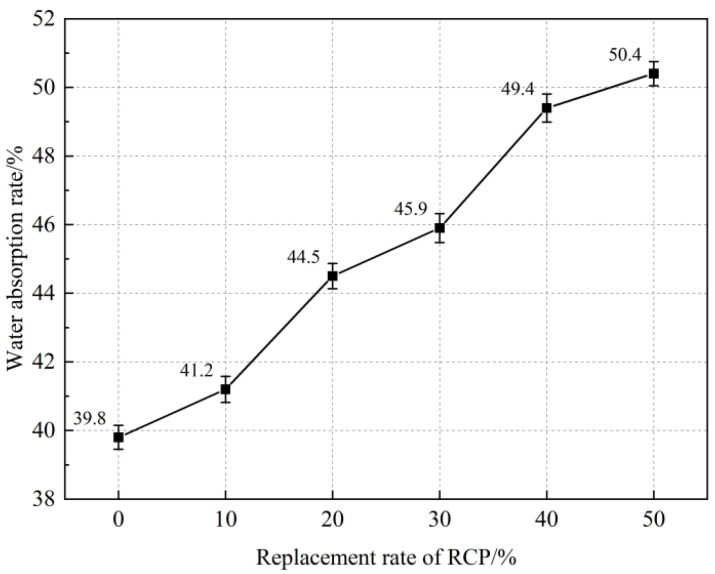
Water absorption of AASR foamed concrete with different RCP Contents.

**Figure 7 materials-16-05728-f007:**
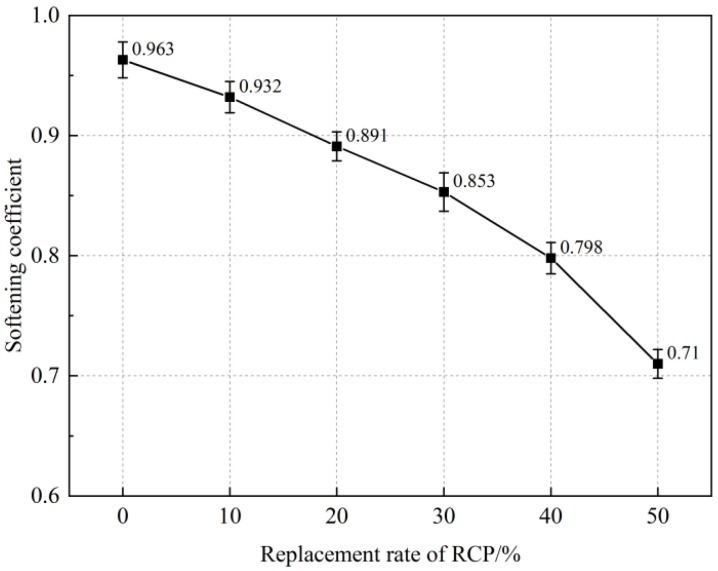
Softening coefficient of AASR foamed concrete with different RCP Contents.

**Figure 8 materials-16-05728-f008:**
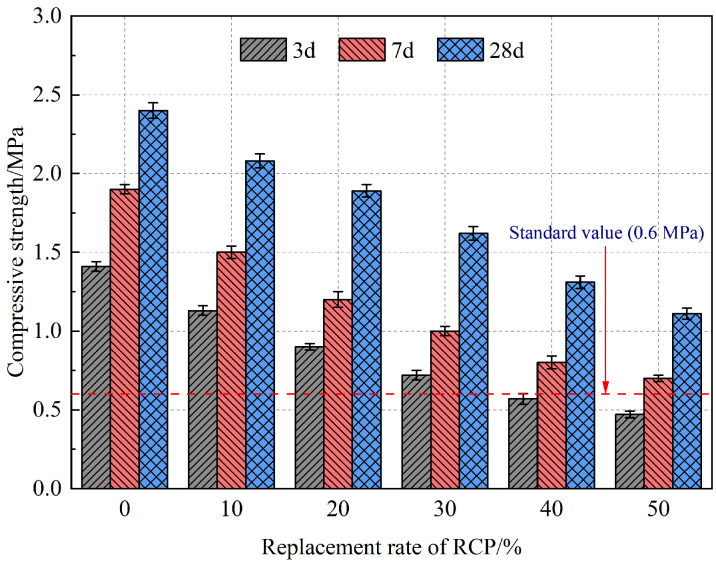
Compressive strength of AASR foamed concrete with different RCP Contents.

**Figure 9 materials-16-05728-f009:**
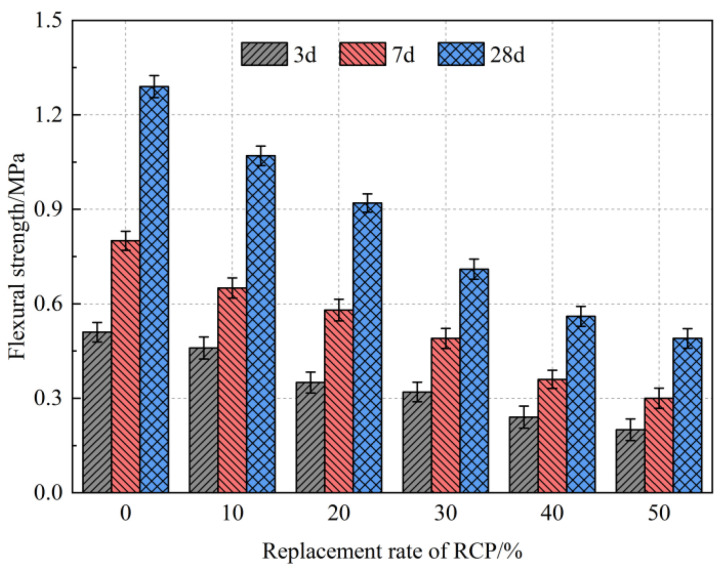
Flexural strength of AASR foamed concrete with different RCP Contents.

**Figure 10 materials-16-05728-f010:**
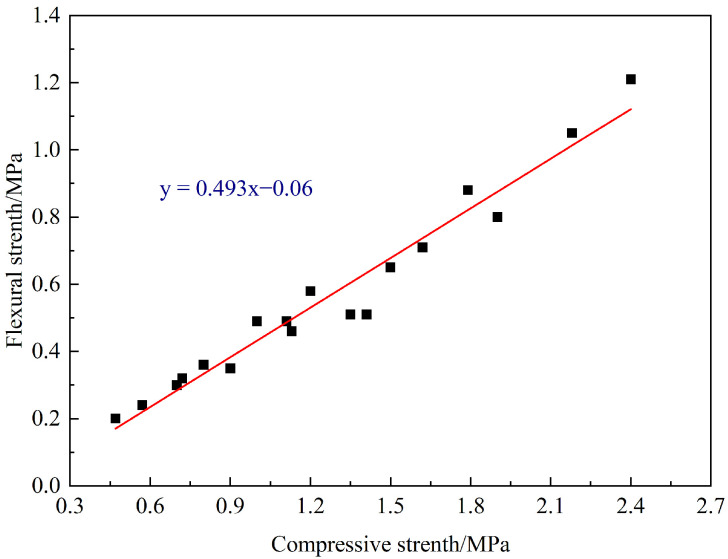
Linear fitting of AASR foamed concrete compressive strength and flexural.

**Figure 11 materials-16-05728-f011:**
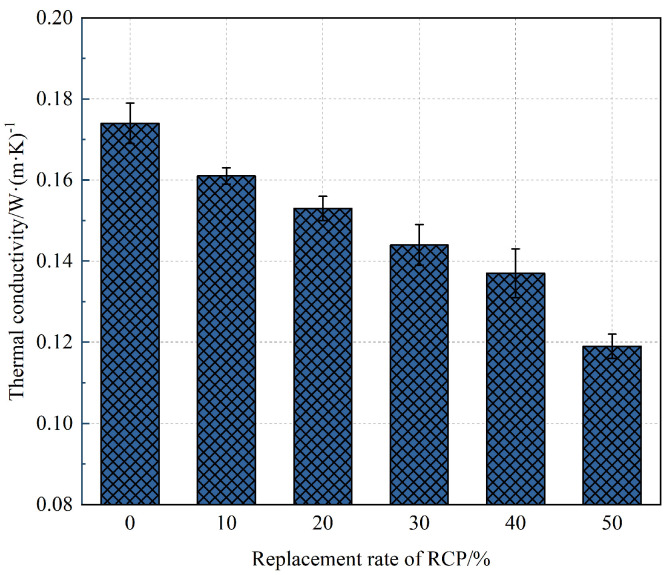
Thermal conductivity of AASR foamed concrete with different RCP Contents.

**Figure 12 materials-16-05728-f012:**
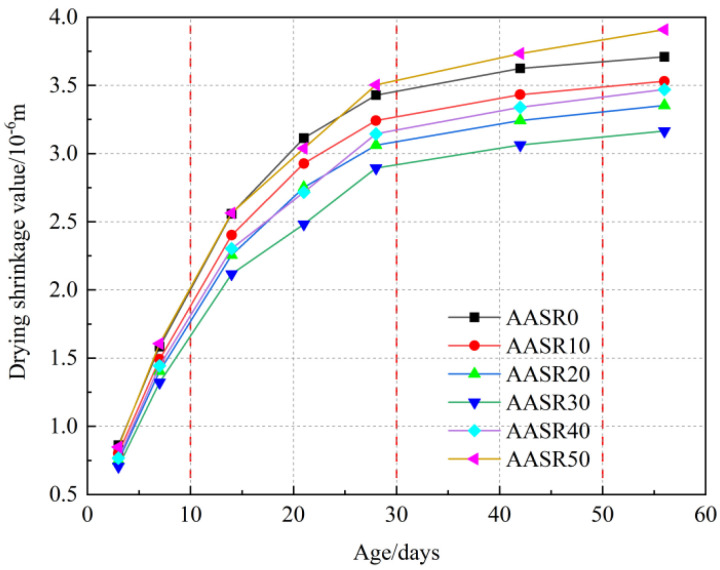
Drying shrinkage of AASR foamed concrete with different RCP Contents.

**Figure 13 materials-16-05728-f013:**
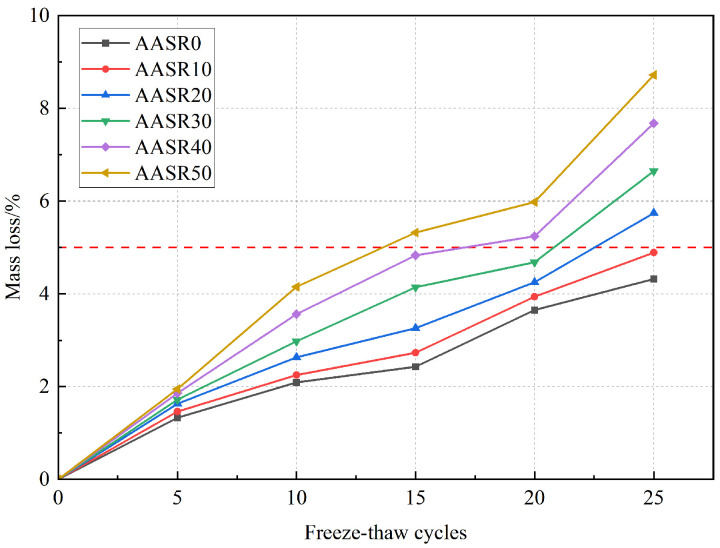
Effect of RCP contents on AASR foamed concrete mass loss under different freeze–thaw cycles.

**Figure 14 materials-16-05728-f014:**
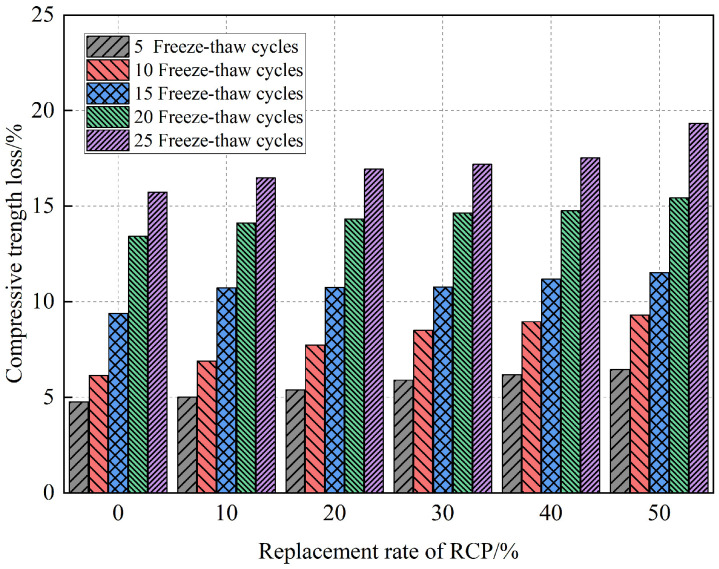
Effect of RCP contents on AASR foamed concrete compressive strength loss under different freeze–thaw cycles.

**Figure 15 materials-16-05728-f015:**
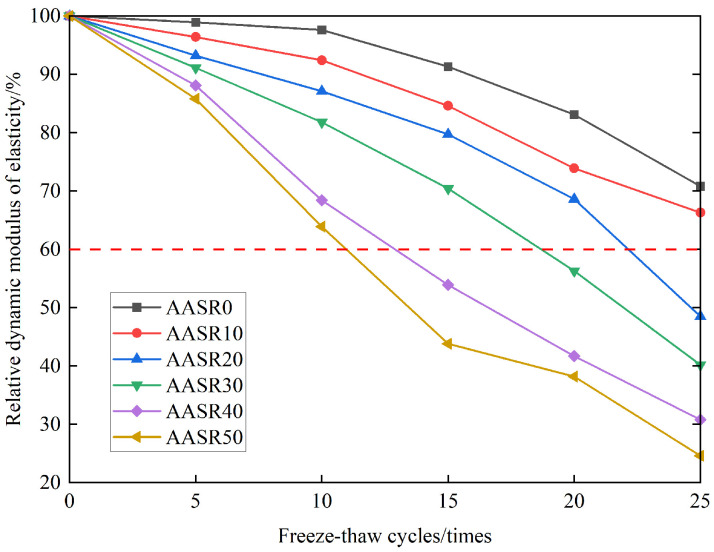
Effect of RCP contents on AASR foamed concrete relative dynamic modulus of elasticity under different freeze–thaw cycles.

**Table 1 materials-16-05728-t001:** Chemical properties of slag and RCP.

Composites	SiO_2_	Al_2_O_3_	CaO	Fe_2_O_3_	MgO	TiO_2_	Na_2_O
Slag	28.7	14.8	38.1	0.42	10.6	1.14	1.78
RCP	40.1	7.45	39.0	3.15	6.18	0.36	0.96

**Table 2 materials-16-05728-t002:** Mix proportions of AASR (kg/m^3^).

Group	Slag	RCP	Alkaline Activator	Water	Foam	Bulk Density
AASR0	375.0	0	85.7	123.7	102.3	546.5
AASR10	341.3	33.7	85.7	123.7	102.3	537.3
AASR20	307.5	67.5	85.7	123.7	102.3	525.2
AASR30	273.7	101.3	85.7	123.7	102.3	510.3
AASR40	240.0	135.0	85.7	123.7	102.3	489.8
AASR50	206.3	168.7	85.7	123.7	102.3	470

## Data Availability

Not applicable.

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
