# Peer review of "Experimental Study on the Application of Recycled Concrete Waste Powder in Alkali-Activated Foamed Concrete"

_materials, 2023, doi:10.3390/ma16175728_

Round 1
Reviewer 1 Report
The submitted manuscript is about experimental study on AASR foamed concrete by adding RCP with different percentage of 0%, 10%, 20%, 30%, 40%, 50% to replace be part of slag. The submitted research seems interesting and can be considered for publication but some points have remained ambiguous and must be discussed. I kindly ask authors to prepare a point-by-point rebuttal response letter and must be subjected to the manuscript as well, considering the following comments:
1) The different percentage of RCP has been used in this study. According to earlier studies, these utilized RCP, is RCF (recycled concrete fine) or RCA (recycled concrete aggregate)? It must be mentioned properly and why different contributions has been implemented? They are going to act as a filler, which are replaced or utilized as coarse or fine aggregates?
2) Were any specific test methods employed to measure the softening coefficient, drying shrinkage, and frost resistance of AASR foamed concrete?
3) What are the potential implications or practical applications of using AASR foamed concrete with different RCP contents?
4) Did the study investigate the long-term durability of AASR foamed concrete with varying RCP contents?
5) Were any specific techniques or measures proposed to mitigate the reduction in mechanical properties caused by the addition of RCP?
Reviewer 2 Report
In the present study, Alkali-activated Foamed Concrete was prepared by partially replacing slag with recycled concrete powder (RCP). Consider the following points for the revision of the paper.
- In paragraph 2.3 it is explained that RPC substitution rates are 10%, 20%, 30%, 40%, and 50%. In abstract it is mentioned as that the content of RPC is 10%, 20%, 30%, 40%, 50%, mass fraction. Choose if the percentage are content of substitution percentage.
- A native English speaker should read the paper. For example, the authors should be careful with the verb tenses (First, mix the mineral powder and RCP evenly in a mixer, then pour water and alkali activator…./ Determine the fluidity, dry density, water absorption…).
- In figure 6 do not use a line if you do not know the relationship between the two parameters. Same in figures 12, 13, 15.
- A native English speaker should read the paper. For example, the authors should be careful with the verb tenses (First, mix the mineral powder and RCP evenly in a mixer, then pour water and alkali activator…./ Determine the fluidity, dry density, water absorption…).
Reviewer 3 Report
Materials-2508628: Experimental study on the application of recycled concrete waste powder in alkali-activated foamed concrete.
General comments
The manuscript reports on an experimental research project whereby recycled concrete waste powder was applied in alkali-activated foamed paste. As apparently no aggregates were used, it is not clear why the authors have referred to the material as “concrete”. The use of powder would appear to suggest that reference to “paste” would be the more appropriate term. In Section 3.3, the word “paste” has been used to refer to the material, and thus it is indeed possible to adopt this throughout the manuscript. Alternatively, the authors could include a brief re-definition of “concrete” as used in the manuscript.
In the abstract, the authors have summarised the tests conducted in the experimental programme reported in the manuscript. However, tests for thermal conductivity and for freeze-thaw effects are not mentioned in the list.
Although the manuscript is well-written in general, there are numerous minor issues of concern, for which an annotated copy is attached with a view to assisting the authors. In addition, the section below on specific comments may also be further help to the authors.
Special comments
1. Table 2. It is not clear how the entries in Column 3 (RCP values) were obtained. It is recommended that these values be re-checked.
2. Paragraph immediately below Table 2. It is not clear which mineral powder is being referred to here. Is this the slag? In the entire description of mixing, it is not clear when or where the slag comes in.
3. Equation 1. The symbol used for density in the formula appears different from the one described in the text.
4. Section 3.3. The meaning of “hole wall” would not be obvious to many readers. Its definition, as well as that of softening coefficient, would be helpful to the reader.
5. Figure 8. The 28-day compressive strength at 50%RCP appears higher than 1.0, yet 0.9 is used in the text in Section 3.4.
6. Section 3.6. The term “hole wall” is used again. See comments in item 4 above.
7. Figure 11. A narrower y-axis scale (as shown in the annotated copy) would magnify the changes in thermal conductivity with replacement rate of RCP.
8. Section 3.7. The entire section could benefit from a re-write, especially to ascertain/confirm that the slow growth and rapid growth are both as stated by the authors.
9. Figure 12. It would be helpful to indicate the three sections/parts described in the text. In addition, an additional or inset graph plotting the data at the age of 56 days would show more clearly the variation of drying shrinkage upon changes in RCP.

As per notes to editor and authors
Round 2
Reviewer 2 Report
First point: think if the term “percentage” is better than “rate”
Second point: Authors did not pay attention to the whole paper. For example, in paragr. 2.3: “According to the standards [30–32], Determine the fluidity, dry density, water absorption, and softening coefficient of AASR foamed concrete were determined.”
Third point: Knowing that some authors use the line graph instead of scatter graph for presenting their experimental results, in this point the reviewer expressed an opinion that this line could mislead the reader for the change rate. For example, in figure 13 it seems that at 20 cycles the mass loss is less than in previous cycles for AASR50, AASR40, AASR30. The same change rate is not for AASR0. Is it a safe way to assume this? A better way is, if possible, a model fitting. It is a better way of expressing the experimental results.
Additionally:
- Tables are not properly formatted so they fit in the page.
- Paragr. 2.2: use “water to binder” instead “water binder”
